



# Observations from the NOAA P-3 aircraft during ATOMIC

Robert Pincus[1,2], Chris W. Fairall[2], Adriana Bailey[3], Haonan Chen[4,2], Patrick Y. Chuang[5], Gijs de Boer[1,2], Graham Feingold[6], Dean Henze[7], Quinn T. Kalen[8], Jan Kazil[1,6], Mason Leandro[5], Ashley Lundry[8], Ken Moran[1,2], Dana A. Naeher[8], David Noone[7,9], Akshar J. Patel[8], Sergio Pezoa[2], Ivan PopStefanija[10], Elizabeth J. Thompson[2], James Warnecke[8], and Paquita Zuidema[11]

[1]Cooperative Institute for Research in Environmental Sciences, University of Colorado, Boulder, Colorado, USA
[2]NOAA Physical Sciences Laboratory, Boulder, Colorado, USA
[3]National Center for Atmospheric Research, Boulder, Colorado, USA
[4]Cooperative Institute for Research in the Atmosphere, Colorado State University, Fort Collins, Colorado, USA
[5]Department of Earth and Planetary Sciences, University of California Santa Cruz, Santa Cruz, California, USA
[6]NOAA Chemical Sciences Laboratory, Boulder, Colorado, USA
[7]College of Earth, Ocean, and Atmospheric Science, Oregon State University, Corvallis, Oregon, USA
[8]NOAA Aircraft Operation Center, Lakeland, Florida, USA
[9]Department of Physics, University of Auckland, New Zealand
[10]Prosensing, Amherst, Massachusetts, USA
[11]Rosenstiel School of Marine and Atmospheric Science, University of Miami, Miami, Florida, USA

**Correspondence:** Robert Pincus (Robert.Pincus@colorado.edu)

**Abstract.** This paper describes observations obtained during the Atlantic Tradewind Ocean-Atmosphere Mesoscale Interaction Campaign (ATOMIC) by the US National Oceanic and Atmospheric Administration's (NOAA) Lockheed WP-3D Orion research aircraft based on the island of Barbados during the period Jan 17 - Feb 11 2020. The aircraft obtained 95 hours of observations over eleven flights, many of which were coordinated with the NOAA research ship R/V Ronald H. Brown and

autonomous platforms deployed from the ship. Each flight contained a mixture of sampling strategies including: high-altitude circles with frequent dropsonde deployment to characterize the large-scale environment; slow descents and ascents to measure the distribution of water vapor and its isotopic composition; stacked legs aimed at sampling the microphysical and thermodynamic state of the boundary layer; and offset straight flight legs for observing clouds and the ocean surface with remote sensing instruments and the thermal structure of the ocean with *in situ* sensors dropped from the plane. The characteristics of the *in*

*situ* observations, expendable devices, and remote sensing instrumentation are described, as is the processing used in deriving estimates of physical quantities. Data archived at the National Center for Environmental Information include flight-level data such as aircraft navigation and basic thermodynamic information (doi:10.25921/7jf5-wv54); high-accuracy measurements of water vapor concentration from an isotope analyzer (doi:10.25921/c5yx-7w29); profiles of sea water temperature made with Airborne eXpendable BathyThermographs (AXBTs, doi:10.25921/pe39-sx75); profiles of radar reflectivity, Doppler velocity,

and spectrum width from a nadir-looking W-band (94 GHz) radar (doi:10.25921/n1hc-dc30); estimates of cloud presence, the cloud top location, and the cloud-top radar reflectivity and temperature, along with estimates of 10-m wind speed obtained from remote sensing instruments operating in the microwave and thermal infrared spectral regions (doi:10.25921/x9q5-9745);





and ocean surface wave characteristics from a Wide Swath Radar Altimeter (doi:10.25921/qm06-qx04). Data are provided as netCDF files following Climate and Forecast conventions.

# 1    Observing the atmosphere and ocean in the winter-time trades

As part of the Atlantic Tradewind Ocean-Atmosphere Mesoscale Interaction Campaign (ATOMIC) the US National Oceanic and Atmospheric Administration (NOAA) operated a Lockheed WP-3D Orion research aircraft from the island of Barbados during the period Jan 17 - Feb 11 2020. The aircraft, known formally as N43RF and informally as "Miss Piggy," is one of two

such aircraft in NOAA's Hurricane Hunter fleet. ATOMIC occurred as part of the field campaign EUREC[4]A (Elucidating the role of clouds-circulation coupling in climate, see Bony et al., 2017) focusing on relationships between oceanic shallow trade cumulus clouds and their environment, including the role of air-sea interactions.

ATOMIC included sampling by a cruise by the NOAA ship Ronald H. Brown (RHB or Ron Brown) and deployments of autonomous aircraft and ocean vehicles. Measurements from the ocean platforms are described in Quinn et al. (2020) and

other forthcoming papers. The main experimental area for EUREC[4]A was just east of Barbados. Both the P-3 and the RHB primarily operated east of the EUREC[4]A area (i.e. east of 57°E), nominally upwind, within the "Tradewind Alley" (see Stevens et al., 2020) extending eastwards from the island of Barbados towards the Northwest Tropical Atlantic Station buoy near 15°N, 51°W. Many of the eleven P-3 flights included excursions to the location of the RHB and sampling of atmospheric and oceanic conditions around the ship and other ocean vehicles. Because of its large size and long endurance (most flights were 8-9 hours

long) the P-3 was tasked with obtaining a wide array of observations including remote sensing of clouds and the ocean surface, *in situ* measurements within clouds and of isotopic composition throughout the lower troposphere, and the deployment of expendable profiling instruments in the atmosphere and ocean.

This paper describes observations made by the P-3 aircraft during ATOMIC. The next section describes the flights during which the measurements were obtained including the flight plans designed to meet each objective. Instrumentation is described

in section 3. Data processing, including the calculation of derived quantities from one or more instruments, is detailed in Sec. 4, which also includes examples and a few comparisons with measurements made by other platforms. Some measurements obtained from the P-3 will be included in cross-experiment data sets to be described elsewhere.

# 2    Sampling strategy

ATOMIC's goals, as the name implies, include illuminating the role of mesoscale circulations in the ocean and atmosphere as

they influence the coupling between the two. As a result the flight strategies included a mix of four different kinds of segments:





1. High-altitude (nominally 24000 ft/7.5 km) circles, nominally of 90 km radius, during which twelve dropsondes (see Sect. 3.2.1) were deployed to characterize the large-scale vertical motion (Bony and Stevens, 2019). Many of the dropsonde circles were centered on the position of the Ron Brown; others were in the location near Barbados that was routinely sampled by the German HALO (High Altitude and LOng range research) aircraft. These were typically the first element of each flight although the three night flights were an exception.

2. Slow descents and ascents to sample thermodynamic profiles and the isotopic composition of water vapor (see Sect. 3.1.2). This pattern was usually flown at the end of the first dropsonde circle, descending from the circle level to 500 ft/150 m above the surface, then ascending to the flight level required for the next pattern.

3. *In situ* cloud sampling patterns, a series of vertically-stacked straight and level legs at altitudes determined during flight. These altitudes were chosen to sample near the ocean surface, just below cloud base, one or more levels within the cloud layer, and just above it. The location of these patterns was determined by the presence and characteristics of the clouds on the flight day.

4. Sets of horizontally-offset long straight legs ("lawnmower patterns") designed to sample the co-variability of clouds and the ocean, emphasizing observations of low-level clouds, ocean temperature profiles (section 3.2.2) and the characteristics of ocean surface waves (section 3.3.2). These patterns were flown at 9000-10000 ft/2.75-3 km so the aircraft could be depressurized to deploy Airborne eXpendable BathyThermographs (AXBTs); this altitude also provides good sensitivity for remote sensing of the ocean surface and clouds. These flight patterns were placed over regions of sea surface temperatures gradients and/or areas being sampled by autonomous ocean vehicles (surface drifters, wave gliders) deployed from the Ron Brown.

Transits between Barbados and the daily operating area offered further opportunities for deploying dropsondes and AXBTs and for remote sensing. The P-3 flew eleven flights during ATOMIC for a total of 95 hours. The first eight flights took place during the day, with nominal take-off times at 13:00 UTC (9:00 local time); the last three took place overnight, with takeoff times between 02:00-03:30 UTC (local times between 22:00 and 23:30 pm the previous day). Table 1 provides an overview of sampling strategies and other information for each flight. A plan (map) view of the flight tracks is shown in Fig. 1; altitudes are shown as a function of flight time in Fig. 2.

## 3  Instrumentation and initial data processing

Table 2 describes the instrumentation on and deployed from the P-3 during ATOMIC. The instrumentation is consistent with that used during hurricane reconnaissance flights and other scientific missions with the addition of the water vapor isotope analyzer provided by the National Center for Atmospheric Research (see Sec. 3.1.2) and the nadir-looking W-band (94 GHz) cloud radar provided by NOAA's Physical Science Lab (Sec. 3.3.1). Many of the basic *in situ* measurements are combined to provide derived quantities (e.g. wind speed, relative humidity) described in Sec. 4.

**Table 1.** Flight sampling strategies employed on each flight by the P-3 during ATOMIC. Flight date is UTC and most flights were 8-9 hours long. Numbers in parenthesis show the number of AXBTs for which valid data was obtained. "RHB" indicates that the R/V Ronald H. Brown was at the center of a dropsonde circle. Most AXBT patterns deployed 20 instruments. "Cloud" indicates the number of cloud patterns flown; each typically involved sampling at four or five altitudes. See also Table 3 of Quinn et al. (2020). Detailed reports from each flight are available at the EUREC⁴A data portal (https://observations.ipsl.fr/aeris/eurec4a/).

| Flight date | Circles | Dropsondes | AXBTs deployed (good) | Cloud patterns | Notes |
|---|---|---|---|---|---|
| 17 Jan | 1 | 23 | | 2 | RHB |
| 19 Jan | 1 | 28 | 40 (37) | | RHB; Second isotope profile on return |
| 23 Jan | 2 | 38 | 40 (38) | | RHB (circle 1) |
| 24 Jan | 2.5 | 16 | | 2 | Coordinated flight with EUREC⁴A - no dropsondes during circles |
| 31 Jan | 1 | 25 | | 2 | RHB |
| 03 Feb | 1 | 22 | 21 (21) | 1 | RHB; First flight with *in situ* microphysics; Early return |
| 04 Feb | | 31 | 20 (20) | | |
| 05 Feb | 1 | 29 | 20 (19) | 3 | |
| 09 Feb | 1 | 32 | 10 (10) | 2 | Night flight, RHB |
| 10 Feb | 1 | 32 | | 2 | Night flight, RHB |
| 11 Feb | | 44 | 15 (15) | | Night flight, RHB fly-by (no circle) |

## 3.1 In-situ measurements

### 3.1.1 Flight level data

Flight level data are recorded every second from the sensors installed on the P-3 via the Airborne Atmospheric Measuring and
Profiling System (AAMPS). Some quantities are measured by multiple sensors; such values are denoted within files prepared by NOAA's Aircraft Operations Center (AOC) with a trailing integer for each independent measurement (e.g. TDM.1, TDM.2 and TDM.3 denote dewpoint temperature measurements from three independent sensors). Flight level data were post-processed and quality controlled by the flight directors (authors QK and AL during ATOMIC) after each flight, typically within a day during the campaign. Each sensor's data is verified to ensure that it represents sound meteorological conditions, then is marked
valid on the QC Checklist included in the Mission Documents (available from https://seb.noaa.gov/pub/acdata/2020/MET/ in directories labeled by flight date and the letter "I" to denote N43RF). In cases where there is more than one reliable sensor, the one sensor is set as the reference, i.e. TDMref. The reference sensor is chosen to minimize data intermittency and maximize both comparisons to independent measurements (e.g. temperature may be compared to dropsondes) and self-consistency among measurements. The intent is to chose a single sensor which best represents the flight overall, even if this sensor might have
periods of bad data (e.g. overshooting by the chilled-mirror dew point sensors) during the flight when other sensors might be more reliable. Additional parameters are derived (variable names end in ".d") and corrected (variable names end in ".c") from

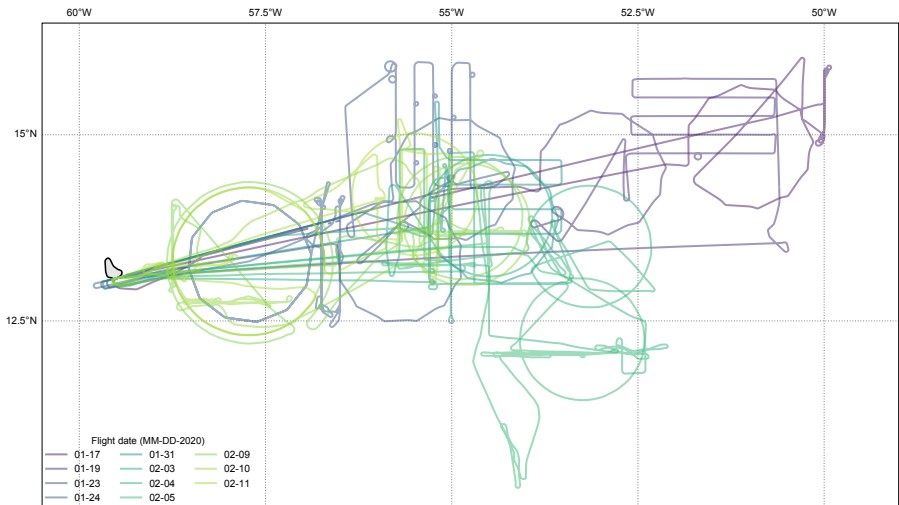

**Figure 1.** Flight tracks for the eleven flights made by the NOAA P-3 aircraft during ATOMIC. Most dropsondes were deployed from regular dodecagons during the first part of the experiment with short turns after each dropsonde providing an off-nadir look at the ocean surface useful for calibrating the W-band radar. A change in pilots midway through the experiment led to dropsondes being deployed from circular flight tracks starting on 31 Jan. AXBTs were deployed in lawnmower patterns (parallel offset legs) with small loops sometimes employed to lengthen the time between AXBT deployment to allow time for data acquisition given the device's slow fall speeds. Profiling and especially *in situ* cloud sampling legs sometimes deviated from straight paths to avoid hazardous weather.

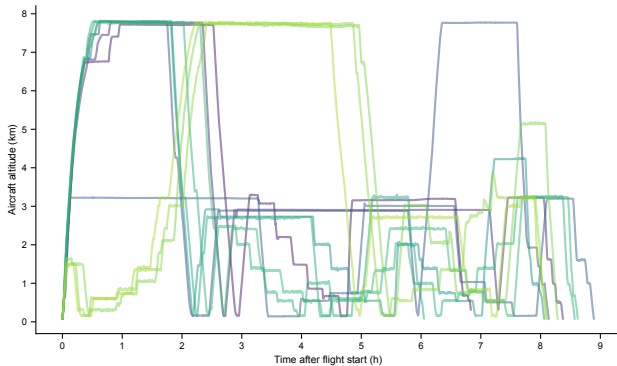

**Figure 2.** Flight altitude as a function of time after take-off for the eleven flights by the NOAA P-3 aircraft during ATOMIC. Sondes were dropped from $\sim 7.5$ km, with each circle taking roughly an hour; transits were frequently performed at this level to conserve fuel. Long intervals near 3 km were used to deploy AXBTs and/or characterize the ocean surface with remote sensing. Stepped legs indicate times devoted to *in situ* cloud sampling. On most flights the aircraft climbed quickly to roughly 7.5 km, partly to deconflict with other aircraft participating in the experiment. On the three night flights, however, no other aircraft were operating at take-off times and cloud sampling was performed first, nearer Barbados than on other flights.





**Table 2.** Instrumentation aboard or deployed from the P-3 aircraft during the ATOMIC field campaign. Most instruments are the same or similar to those used during hurricane reconnaissance and other scientific missions, though the water vapor isotope analyzer and W-band (94 GHz) radar were deployed specifically for this field campaign. See also Table A5 in Stevens et al. (2020).

| Instrument/sensor/package | Measurand | Notes |
|---|---|---|
| *In situ* measurements | | |
| NovAtel DL-V3 GPS | Aircraft location, orientation | Primary (GPS.3, see Sec. 3.1.1) |
| Northup Grumman RINU-G | Aircraft heading | accuracy $\pm 0.02°$ |
| Rosemount 1281AF2B2BEP3 | Static pressure | accuracy $\pm 1.6$hPa |
| Rosemount 102AL | Air temperature | accuracy $\pm 0.1°$C |
| Buck Research 1011C | Dewpoint temperature | accuracy $\pm 0.5°$C; TDM.1 (see Sec. 3.1.1) |
| EdgeTech Vigilant | Dewpoint temperature | accuracy $\pm 0.5°$C; TDM.2 (see Sec. 3.1.1) |
| Vaisala PTB 220 | Cabin pressure | |
| Water vapor isotope analyzer measurements | | |
| Picarro L2130-i | Water vapor concentration, isotopic composition | |
| Microphysics measurements | | |
| Cloud and Aerosol Spectrometer | Individual particle size | 0.5-50 μm |
| Cloud Droplet Probe | Individual particle size | 2-50 μm (not functional during ATOMIC) |
| Cloud Imaging Probe | Individual particle size | 25-1550 μm |
| Precipitation Imaging Probe | Individual particle size | 100-6200 μm |
| Expendables | | |
| Vaisala RD41 dropsondes | Temperature, humidity, pressure, position vs. altitude | |
| Lockheed Martin Sippican AXBTs | Sea water temperature vs. depth | |
| Remote sensors | | |
| Heitronics KT19.85II | Infrared radiation 9.6 - 11.5 μm | One side-looking, one down-looking |
| | $\pm 0.5°$C plus 0.7% of the difference between target and housing temperatures | |
| W-band radar | Intensity vs. Doppler shift | NOAA Physical Sciences Lab |
| WSRA | 16 GHz radar reflectivity | Prosensing |
| SFMR | C-band brightness temperatures | Prosensing |

these data. AOC produces and distributes one netCDF file per flight. Both raw data and the AOC summary file are available for all flights from NOAA's National Center for Environmental Information.





### 3.1.2  Water vapor stable isotope analyzer

During ATOMIC the P-3 was equipped with a flight-ready Picarro L2130-i water vapor isotopic analyzer which measured the concentration of water vapor and its isotopic composition at 5 Hz frequency. The isotope ratio measurements, which are part of a broad suite of such observations made during EUREC[4]A, are reported elsewhere; here we describe the instrument's fast and accurate measurements of water vapor concentration (i.e. mixing ratio). ATOMIC was the first flight campaign for this newly-developed instrument.

While in flight, the isotopic analyzer drew in ambient air through a backwards-facing 0.25-in copper tube centered within a National Center for Atmospheric Research HIAPER Modular Inlet (HIMIL). This ensured the selective sampling of water vapor (versus total water). Because mass but not volumetric flow was controlled through the copper tubing, the time delay ($\tau$ in seconds) for air entering the HIMIL to reach the isotopic analyzer varied as a function of pressure and temperature.This delay may be approximated as $\tau = 1.0748 p/T_{set}$ where $T_{set}$ is the set point (K) of the heaters wrapping the copper tubing inside the

aircraft cabin, $p$ is the ambient pressure (hPa) recorded by the aircraft, and the constant (units of $\mathrm{s\,K/hPa}$) represents both the best approximation for the inner volume of the copper tube, including the 6 ft inside the cabin and 1 ft extending out through the HIMIL pylon, and the scale factor required to relate $T_{set}$ and $p$ to the standard conditions under which the volumetric flow rate of the isotopic analyzer is known. The tube inside the cabin was heated to 313.15 K during the first two flights and 321.15 K thereafter, resulting in a typical time delay of $3.4 \pm 0.3$ s near the surface, which reduces by approximately 1 s for every 300

hPa gained in altitude. Not all parameters in this equation are well constrained or fully representative of the exact sampling conditions within the inlet; the uncertainty in $\tau$ may be roughly estimated by considering $p \pm 75$ hPa.

Mixing of water vapor within the inlet system, and with molecules that have adsorbed to the copper tubing, also partially smooths high frequency signals. These effects are, however, fairly small and consistent across flights. The time response of the aircraft's chilled mirror hygrometer, in contrast, is quite variable and depends on flight conditions, and the hygrometer is subject

to both overshooting (e.g. when the measured signal surpasses the expected value following a rapid rise in environmental water vapor concentration) and ringing (i.e. rapid oscillations around the expected value) during rapid and large changes in water vapor concentration. These features can be seen in Fig. 3 which also highlights the hygrometer's much slower time response (as compared to the isotopic analyzer) in the low humidity conditions found at the highest flight altitudes. Outside of these time periods, the agreement between the hygrometer and isotopic analyzer is quite good (lower panel). Given the more consistently

accurate measurements of the isotopic analyzer during ATOMIC we encourage its use in preference to the aircraft hygrometer for characterizing the thermodynamic state of the atmosphere.

### 3.1.3  Microphysics

A number of instruments for measuring aerosol and hydrometeor microphysical properties were onboard the P-3 during ATOMIC. All are standard instrumentation manufactured by Droplet Measurement Technologies. For measuring aerosol and

cloud drops, a Cloud and Aerosol Spectrometer (CAS; nominal diameter range 0.5 to 50 μm) and a Cloud Droplet Probe (CDP; nominal diameter range 2 to 50 μm) were both deployed. For precipitation drops, a Cloud Imaging Probe (CIP; nominal




diameter range 25 μm to 1.55 mm) and a Precipitation Imaging Probe (PIP; nominal diameter range 100 μm to 6.2 mm) were deployed. The operating principles for these instruments are described in the survey by Baumgardner et al. (2011). All instruments were factory calibrated immediately before the project.

Microphysics measurements were not made during all ATOMIC flights. The computer controlling the microphysical instruments failed on the first flight and took some time to replace, so that no microphysical measurements were made during the first four flights. The CDP never functioned properly during the experiment, while the CIP and PIP did not properly function until the sixth flight on 03 Feb 2020. Measurements are available from all other instruments for all remaining flights up to the end of the project.

## 3.2    Expendable instrumentation

### 3.2.1    Dropsondes

The P-3 released 320 Vaisala Dropsonde RD41s during ATOMIC at the locations shown in Fig. 4. Most were released from 24000 ft/7.5 km though some were released from slightly lower altitudes during transits and others from 9000-10000 ft/2.75-3 km during cloud and AXBT flight patterns. The RD41 sensors measure pressure, temperature, and humidity as the package
falls from the plane, slowed by a parachute (Hock and Franklin, 1999). A GPS package provides location from which wind direction and wind speed are calculated and reported in real time. Measurements are available from the aircraft flight level to the ocean surface. Dropsondes from the P-3 were processed in real time during flight and made available for assimilation over the Global Telecommunications System.

### 3.2.2    AXBTs

A total of 165 AXBT instruments were deployed from the P-3 over seven flights at locations shown in Figure 5. Most were released at or near 9000 ft /2.75 km. The AXBTs, manufactured by Lockheed Martin Sippican, collect ocean temperature as a function of time after launch. AXBTs normally begin transmitting data when the sensor enters the ocean. One file was produced for each AXBT sensor by removing any extraneous observations obtained before splashdown, then converting time to depth using an in-water fall speed of $1.594\ \mathrm{m\,s^{-1}}$. Location is determined from the aircraft navigational information at the time the
AXBT was released. A median filter was applied to remove most (but not all) spurious outliers in ocean water temperature.

## 3.3    Remote sensing observations

### 3.3.1    Physical Sciences Laboratory W-band radar

Remote sensing instrumentation on the P-3 during ATOMIC included the NOAA Physical Sciences Lab (PSL) W-band (94-GHz) pulsed Doppler radar. The hardware and processing are described in Moran et al. (2012). It has been deployed from the
surface (ships and land stations) looking up and from NOAA P-3 aircraft looking down. During ATOMIC the airborne radar





was operated with 220 30-m range gates with a dwell time of 0.5 s. The minimum detectable reflectivity was -36 dBZ at a range of 1 km although accurate estimates of Doppler properties require about -30 dBZ at 1 km.

Radar data is post-processed following Fairall et al. (2018). Standard processing produces vertical profiles of estimates of reflectivity, Doppler velocity, spectrum width, and cloud-free signal-to-noise ratio, converted to a uniform grid referenced to
the sea surface rather than as distance from the aircraft. Reflectivity profiles are corrected for attenuation by atmospheric gases and precipitation. Absorption by water vapor and oxygen is calculated based on temperature, pressure, and relative humidity profiles measured by dropsondes using the model suggested by Union (2013). Attenuation by precipitation is estimated using inversions and relationships from Hitschfeld and Bordan (1954) following Iguchi and Meneghini (1994). Measured Doppler velocity is corrected for the pitch and roll components of aircraft motion. The vertical speed of the aircraft is calculated from
flight level data (Sec. 3.1.1) and taken into account in the Doppler velocity correction, especially during aircraft ascents and descents. An example from an hour of flight (18-19 UTC on Jun 19) is shown in Fig. 6.

### 3.3.2 Wide Swath Radar Altimeter

The NOAA Wide Swath Radar Altimeter (WSRA, see Walsh et al., 2014; PopStefanija et al., 2020) is a digital beam-forming radar altimeter operating at 16 GHz in the Ku band. It generates 80 narrow beams spread over $\pm 30°$ to produce a topographic
map of the sea surface waves and their backscattered power. These measurements allow for continuous reporting of directional ocean wave spectra and quantities derived from this including significant wave height, sea surface mean square slope, and the height, wavelength, and direction of propagation of primary and secondary wave fields. Rainfall rate is estimated from path-integrated attenuation.

### 3.3.3 Stepped Frequency Microwave Radiometer

The Stepped Frequency Microwave Radiometer (SFMR) is a nadir-looking microwave radiometer built by ProSensing, Inc. of Amherst, MA, USA. The instrument measures brightness temperatures of the ocean surface and intervening atmosphere at six C-band (4-7 GHz) frequencies. Surface wind speed and average columnar rain rate can be inferred from these brightness temperature values (Uhlhorn et al., 2007). Since its initial deployment in 1980 the hardware and retrieval methods have been improved several times; Sapp et al. (2019) describe the most recent update.

### 3.3.4 Infrared radiometers

The P-3 deployed three Heitronics KT19.85 passive infrared radiometers which measure radiation in the 9.6 - 11.5 μm spectral range. One radiometer points horizontally out the port side of the plane; the others are zenith- nadir-looking. Measured radiation is converted to a brightness temperature. This system has a resolution of 0.1°C and an accuracy of 0.5°C plus 0.7% of the difference between the target and instrument housing temperatures. The field of view of 0.5° and the response time
approximately 0.5 s.





**Table 3.** Data available from the P-3 during ATOMIC. Data is packaged as one file per type per flight day. Subsections within Sec. 4 describe the production of data beyond routine flight level data (Sec. 3.1.1 and the routinely-processed radar observations (Sec. 3.3.1).

| File type | Reference | Freq. | Data provided |
|---|---|---|---|
| W-band radar | Sec. 3.3.1 | 2 Hz | Radar reflectivity, Doppler velocity, spectrum width, signal-to-noise |
| Flight level data | Sec. 4.1.1 | 1 Hz | Raw measurements as described in Table 2 |
| | | | relative humidity |
| | | | aircraft ground speed; true air speed; course over ground; true heading |
| | | | wind speed and direction, wind velocity components ($u$, $v$, $w$) |
| | | | 10-m wind speed and rain rate from SFMR |
| Isotope analyzer water vapor | Sec. 4.1.2 | 1, 5 Hz | Volume and mass mixing ratios and standard errors; relative humidity |
| AXBTs | Sec. 4.2.2 | | sea water temperature (profile) |
| Remote sensing | Sec. 4.3.1 | 2 Hz | ocean temperature estimate from IR measurements |
| | | | Radar and infrared cloud indexes |
| | | | cloud top altitude, wind speed, air temperature, radar reflectivity and Doppler velocity |
| | | | infrared temperature at cloud top |
| | | | rain rate; normalized radar cross section; sea surface mean square slope (W-band) |
| | | | corrected and uncorrected 10-m wind from SFMR |
| WSRA | Sec. 4.3.2 | 0.02 Hz | Directional wave spectrum; peak spectral variance; rainfall rate |
| | | | dominant and secondary wave height, direction, wavelength |
| | | | sea surface mean square slope; significant wave height |

# 4 Post-processed and derived quantities

Data obtained during the experiment have been post-processed and a variety of derived quantities, as described in this section, have been produced. Data files are organized topically, as described in Table 3 and explained more fully in this section. One file per day is provided for each of the entries in the table.

## 4.1 *In situ data*

### 4.1.1 Flight level data

To simplify analysis and ease comparisons to other observations we have produced modestly reformatted data files containing a subset of flight level data. These files contain only the reference value of quantities measured by multiple sensors. Some variables are re-named for consistency with other platforms in ATOMIC and/or EUREC[4]A. Metadata are added or otherwise 195 made consistent with conventions developed for the experiments.





### 4.1.2 Isotope analyzer

Water vapor measurements from the isotopic analyzer are proportional to the ratio of the moles of water vapor to the moles of moist air (dry air plus water vapor) and are reported as a volume mixing ratio in parts per million volume (ppmv). These are provided alongside estimates of the mass mixing ratio – the ratio of the mass of water vapor to the mass of dry air – at both

the analyzer's native time resolution of nominal 5 Hz frequency and at a reduced resolution of 1Hz aligned, through boxcar averaging, with the P-3 aircraft data system. For convenience, estimates of relative humidity are also provided, obtained by multiplying the 1Hz volume mixing ratios by the aircraft static (ambient) pressure measurements and dividing by the saturation vapor pressure, estimated from the aircraft ambient temperature following Hardy (1998). The files are aligned in time with the flight level data and contain no geo-location information, which should instead be extracted from the corresponding flight level

file.

The isotopic analyzer's water vapor measurements have been corrected for a low bias of increasing magnitude at concentrations exceeding 10,000 ppmv, identified using a LiCOR 610 dew point generator. The uncertainty associated with this correction spans 26 to 29 ppmv for the humidity range 200 to 30,000 ppmv but reduces to 12 ppmv upon averaging to 1 Hz. The accuracy of volume mixing ratios below 200 ppmv is unverified.

### 4.1.3 Microphysics

Microphysical data must be heavily post-processed before use since the instruments record a time series of individual particle events. This post-processing is ongoing but not yet complete. The CAS is processed with standard codes available from the manufacturer. (Had the CDP produced useful data during ATOMIC the processing stream would have been similar.) The CIP and PIP provide quick-look data that are qualitatively useful. However, accurate quantitative data requires specialized

processing of the individual particle images. The System for OAP Data Analysis version 2 (SODA-2, https://github.com/abansemer/soda2) is used to process the images to produce drop size distributions from both instruments. There will be two primary final data products. The first is an aerosol size distribution that uses only CAPS measurements and spans 0.5 to 2 μm. The second is a merged cloud drop size distribution that combines CAPS, CIP and PIP instruments and spans 2 μm to 6.2 mm diameter. Preliminary analysis shows good agreement in the overlap region between CAPS and CIP, and CIP and PIP. Integral

properties of these size distributions (e.g. total number concentration, liquid water content) will also be included. All primary products will be reported at 1 Hz. Secondary data products will include the individual size distributions from each instrument (i.e. one distribution each from CAPS, CIP and PIP). Additionally, there will be quick look videos made available in the data archive which will permit anyone to preview the two size distributions as a function of time without doing their own analysis. These videos will also include leg-averaged aerosol and cloud drop size distributions, in addition to basic aircraft information

such as location (latitude-longitude) and altitude.





### 4.2 Expendables

#### 4.2.1 Dropsondes

Observations from the dropsondes (sec. 3.2.1) deployed by the P-3 are being processed alongside similar observations made from the high-altitude HALO aircraft as part of EUREC[4]A. A combined dataset will be made available as part of the Joint dropsonde-Observations of the Atmosphere in tropical North atlaNtic large-scale Environments (JOANNE) described elsewhere. This effort is the dropsonde complement of the compilation of radiosonde observations made during ATOMIC and EUREC[4]A that is described in Stephan et al. (2020) and available at https://doi.org/10.25326/62.

Figure 7 compares dropsonde profiles around the perimeter of the circle centered on the position of the Ron Brown with a radiosonde launched from the ship. The P-3 entered the circle at 15:26 UTC and exited at 16:25, dropping sondes evenly throughout this window; the radiosonde was launched by the ship at 14:43 UTC to meet the synoptic deadline of 16:00 UTC. Despite this small temporal mismatch the thermal structures observed by the radio- and drop-sondes are similar, with a small inversion near 6 km and a larger inversion near 2.5 km, the height of which varies across and around the circle. Large jumps in the moisture field associated with these inversions exhibit similar vertical variability.

#### 4.2.2 AXBTs

Following the processing of dropsonde data for JOANNE we have produced a single file containing all AXBTs profiles obtained during the ATOMIC, interpolated to a standard depth grid at 0.1 m vertical resolution. Figure 8 shows an example from the flight on Jan 19 2020 in which 40 AXBTs were deployed in a lawnmower pattern bracketing five Surface Wave Instrument Floats with Tracking (SWIFT) buoys (Thomson, 2012) deployed from the R/V Ronald H. Brown (see the upper right corner of Fig. 5). Figure 8 shows the ocean temperature as measured by the AXBTs between the surface and 150 m depth, with a near-isothermal mixed layer extending tens of meters and the cooler ocean below 60-80 m. The inset compares the temperature in the three few meters with measurements made by the SWIFT buoys (see Quinn et al., 2020)), nominally at 0.3-0.5 m depth depending on the particular buoy. Upper ocean temperatures measured by the AXBTs span one K; the range across the SWIFTs, which were more geographically confined, is about a fifth of this.

### 4.3 Clouds, rain, and sea state parameters derived from remote sensing

#### 4.3.1 W-band radar and infrared radiometer: clouds, precipitation, and sea state

We use observations from the W-band radar (Sect. 3.3.1) to estimate ocean surface parameters and, in combination with measurements from the downward-looking infrared radiometer (Sect. 3.3.4), to provide estimates of cloud properties. Both sets of parameters are distributed with navigation data interpolated to the 2 Hz radar time base. The file also contains values of the 10 m SFMR wind speed, both as reported by the instrument and as corrected via linear regression using dropsonde winds as the reference.



Estimates of sea state and precipitation rate make use of the strong reflection of the W-band radar from the ocean surface. Following Fairall et al. (2018) we report the measured normalized radar cross section $NRCS_m$ based on the observed reflectivity factor of the ocean surface $dBZ_e(0)$

$$NRCS_m = dBZ_e(0) + 10\log_{10}(\pi^5|K^2|\delta R/\lambda^4) - 180 + dBZ_{attn} \qquad (1)$$

For the PSL W-band with its 30 m range resolution, the second term on the right-hand side of (1) is 137.9, while the correction factor for attenuation by water vapor and oxygen is roughly $dBZ_{attn} = 4$ for typical ATOMIC conditions with the aircraft at 3 km altitude. During ATOMIC the signal from the surface was strong enough to cause some saturation of the receiver, reducing the sensitivity the nearer the aircraft was to the surface. Values of $NRCS_m$ have been further adjusted for this effect as a function of pressure; the correction is small for altitudes higher than 5 km but is as large as 8 dB at 1 km above sea level.

The back-scattered radar return from the ocean surface $\sigma$ depends on both wind speed and viewing angle $\theta$; this dependence can be exploited to estimate the mean square slope $\overline{s^2}$ of surface waves (satellite-borne radar scatterometer wind estimates exploit the same physics). The dependence is usually represented (Walsh et al., 1998; Li et al., 2005) as

$$\sigma = \frac{\Gamma^2}{\overline{s^2} * \cos^4\theta} \exp\left(\frac{-\tan^2\theta}{\overline{s^2}}\right) \qquad (2)$$

where $\Gamma^2$ is a wavelength-dependent constant with value 0.32 at W-band radar frequencies and the theoretical or calculated
normalized radar cross section $NRCS_c = 10\log_{10}(\sigma)$. We solve (2) for $\overline{s^2}$, using observations made a nadir viewing angles ($\theta = 0$) and assuming $NRCS_c = NRCS_m$. These estimates rely on absolute radar calibration.

Following Fairall et al. (2018) column-mean rain rate is determined from the W-band radar using the vertical gradient of radar reflectivity during light rain and the path-integrated attenuation during the infrequent heavy rain observed during ATOMIC.

Clouds are detectable in both the radar reflectivity profile and the observed infrared brightness temperature. A radar cloud
presence index $C^{\text{radar}}$ is determined by examining the maximum signal-to-noise ratio (SNR) within each radar column (excluding the surface return). The clear-sky signal-to-noise level of the radar is nominally -20 dB but a value of about -15 is needed to ensure a valid cloud return. We define the index as $C^{\text{radar}} = \max(\text{SNR}) + 14$ so that values of $C^{\text{radar}} > 0$ indicate clouds. When clouds are detected cloud-top height $z_{ct}$ is estimated as the level closest to the aircraft at which SNR $> 14$. Cloud-top height diagnosed in this way is shown in the middle panel of Fig. 6. We report the wind speed $w(z_{ct})$ and air temperature $T(z_{ct})$ at
this height as determined from daily-mean *in situ* aircraft profiles (not dropsondes). We also report the radar reflectivity factor and Doppler velocity at this height as simple indicators of cloud thickness and the presence of precipitation.

An infrared cloud presence index $C^{\text{IR}}$ is produced based on the observed nadir-looking brightness temperature $T^{\text{IR}}(p)$ made at aircraft operating pressure $p$. We compare clear-sky measurements of $T^{\text{IR}}(p)$ to the near-surface radiometric temperature, determined from the time-mean of infrared radiometer measurements during flight legs at 150 m, to develop a correction
term $\Delta T^{\text{IR}}(p - p') = \text{SST} - T^{\text{IR}}_{\text{clear}}(p - p_{\text{sfc}})$ as a quadratic function of $p - p'$. The infrared cloud index $C^{\text{IR}}$ is defined as the difference between the observed infrared temperature and the value expected in the absence of clouds, i.e. $C^{\text{IR}} = T^{\text{IR}}(p) - (SST - \Delta T^{\text{IR}}(p - p_{\text{sfc}}))$, so that values of $C^{\text{IR}} < 0$ indicate clouds.

The two cloud indexes complement one another. The W-band radar sensitivity is limited and, particularly when the aircraft was transiting or dropping sondes at 7.5 km altitude (about a quarter of the total flight time), many clouds near the surface were





beyond the viewing range from the radar are therefore not detectable in the radar return. For these flight legs the IR cloud index is likely a better indicator of cloudiness. When the aircraft is at or below about 3 km, the radar is very sensitive to clouds and likely detects all clouds with radar reflectivity factor $Z_e > -35$dBZ. Under these circumstances the radar and infrared cloud indexes are quite consistent with one another, as shown for an example hour of observations on 19 January 2020 in Fig. 9.

When cloud top height $z_{ct}$ is available from the radar we use this information to identify cloud-top pressure $p_{ct}$ and, from
aircraft soundings, the temperature at that pressure $T_{ct}^{air}$. This can be compared to IR cloud top temperature corrected for the intervening atmosphere $T_{ct}^{IR} = T^{IR}(p) + \Delta T^{IR}(p - p_{ct})$. Values of $T_{ct}^{IR} \approx T_{ct}^{air}$ indicate that optically-thick clouds fill the infrared radiometer's field of view. In Fig. 10 we show the difference in cloud-top air temperature and apparent IR cloud-top temperature as a function of W-band reflectivity from January 19. Values of $T_{ct}^{IR} - T_{ct}^{air}$ less than zero indicate the cloud is optically thin or does not completely fill the 0.5 s sample and some radiation from the warm sea surface is adding to the measured IR.

#### 4.3.2 WSRA: Sea state and rain rate information

ProSensing Inc. processes raw data from the Wide-Swath Radar Altimeter (WSRA, see sec. 3.3.2) to produce information about the wave state of the ocean surface. Most observations are reported every 50 seconds. These include the power spectrum of surface waves as a function of wavenumber in the north-south and east-west directions; the direction, height, and wavelength of the two most dominant waves; peak spectral variance; and the significant wave height. A plan view of observations obtained
on 19 Jan 2020 is shown in Fig. 11. Column-mean rain rate and surface wave $\overline{s^2}$ are reported every ten seconds. The latter is computed from the decrease of the intensity of the return with scan angle following (2) so, unlike estimate of $\overline{s^2}$ from the W-band radar, estimates from the WSRA do not depend on absolute calibration.

Files with these estimates also contain bookkeeping information (processing parameters and ancillary data) such as aircraft navigation and orientation and other fields that may be useful. In particular, the directional wave spectra calculated from data
collected with WSRA inherently contain a 180° ambiguity of the wave propagation which can generally be eliminated using a rough estimate of a predicted dominant ocean wave direction at the location of the observation point. During ATOMIC the prevailing wind direction (typically ENE to WSW) was used as the predicted ocean wave direction for the entire duration of each flight mission. For completeness WSRA files contain directional wave spectra with and without this ambiguity removed.

During ATOMIC the aircraft operated in a number of modes that were unfavorable for collecting WSRA data. Data should
not be used if the aircraft altitude is less than 500 m or greater than 4000 m, or when the aircraft's pitch or roll exceeds $\pm 3°$. We also recommend using observations only when the peak spectral value is in the range $0.0002 - 0.006$ m$^2$.

### 5  Data preparation for wide dissemination

Not all observations obtained by the P-3 are available at this time. Uniformly-processed dropsonde data will be included in the JOANNE dataset to be described elsewhere. Similarly, isotope ratios from the isotope analyzer aboard the P-3 (sec. 3.1.2)
will be described as part of a paper describing all aircraft-, ship-, and ground-based observations made during the experiments. Measurements from the *in situ* microphysical probes (sec. 3.1.3) will be documented after they have been produced.





**Table 4.** Data described in this paper and archived at NOAA's National Center for Environmental Information. The contents of each file are summarized in Table 3, Document object identifiers (DOIs) point to netCDF files following Climate and Forecast conventions.

| File type | DOI |
| --- | --- |
| W-band radar | 10.25921/n1hc-dc30 |
| Flight level data | 10.25921/7jf5-wv54 |
| Isotope analyzer water vapor | 10.25921/c5yx-7w29 |
| AXBTs | 10.25921/pe39-sx75 |
| Clouds, rain, and sea state from remote sensing | 10.25921/x9q5-9745 |
| WSRA | 10.25921/qm06-qx04 |

Data have been reformatted into netCDF files following CF (Climate and Forecast) conventions (https://cfconventions.org) which provide for units and standard names for variables, a uniform handling of time, and other metadata intended to promote interoperability and interpretability. The files also contain some provenance information following guidance developed for
EUREC[4]A.

## 6   Data availability

Data have been archived at NOAA's National Center for Environmental Information, as detailed in Table 4. This archive represents the version of record. The data are also replicated at the French AERIS data center alongside the wide array of other data from the EUREC[4]A experiment (OpenDAP access via https://observations.ipsl.fr/thredds/catalog/EUREC4A/catalog.html).
NOAA's Physical Sciences Laboratory has also replicated data from ATOMIC along with variants including single profiles from AXBT on the native grid, raw flight level data as supplied by NOAA's Aircraft Operations Center, and hourly radar reflectivity files that may be prove more useful to some users than the archived files containing the entire flight. Data at PSL is available via OpenDAP (https://psl.noaa.gov/thredds/catalog/Datasets/ATOMIC/data/p3/catalog.html) and for download via ftp (ftp://ftp2.psl.noaa.gov/Projects/ATOMIC/data/p3).
Code used to generate the figures in this paper is available at http://github.com/RobertPincus/atomic-p3-data-paper/ (doi to be provided on publication).

*Author contributions.* CWF was the Principal Investigator for the aircraft during the ATOMIC mission. CWF, RP, JK, GF, and PZ each acted as a flight scientist for one or more research flights. AB and DH were responsible for operating the Picarro instrument. GB operated the W-band radar for one of two research flights for which he was onboard and assisted with data processing and conversion to NetCDF. IP
obtained and post-processed data from the WSRA and is the point of contact for data from this instrument. RP coordinated data archiving activities and the production of this manuscript. All authors contributed to the collection and/or processing of the data described.



*Competing interests.* Author GB is a guest editor for the special issue to which this manuscript is submitted. The authors declare that they have no other conflicts of interest.

*Acknowledgements.* We are grateful to the pilots, crew, and ground support staff of the WP-3D for their help in obtaining these measurements.
The ATOMIC field campaign was supported by the NOAA Climate Program Office under the Climate Variability and Predictability Program; collection of data from the P-3 was supported by award GC19-302a. Additional support for processing and collection of ATOMIC WP-3D data sets was provided by the NOAA Physical Sciences Laboratory. Measurements with the isotope analyzer were supported by the National Center for Atmospheric Research, which is a major facility sponsored by the National Science Foundation under Cooperative Agreement 1852977. The authors thank Tom Boyer and John Relph at NOAA's National Center for Environmental Information for essential help in
archiving the data.



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



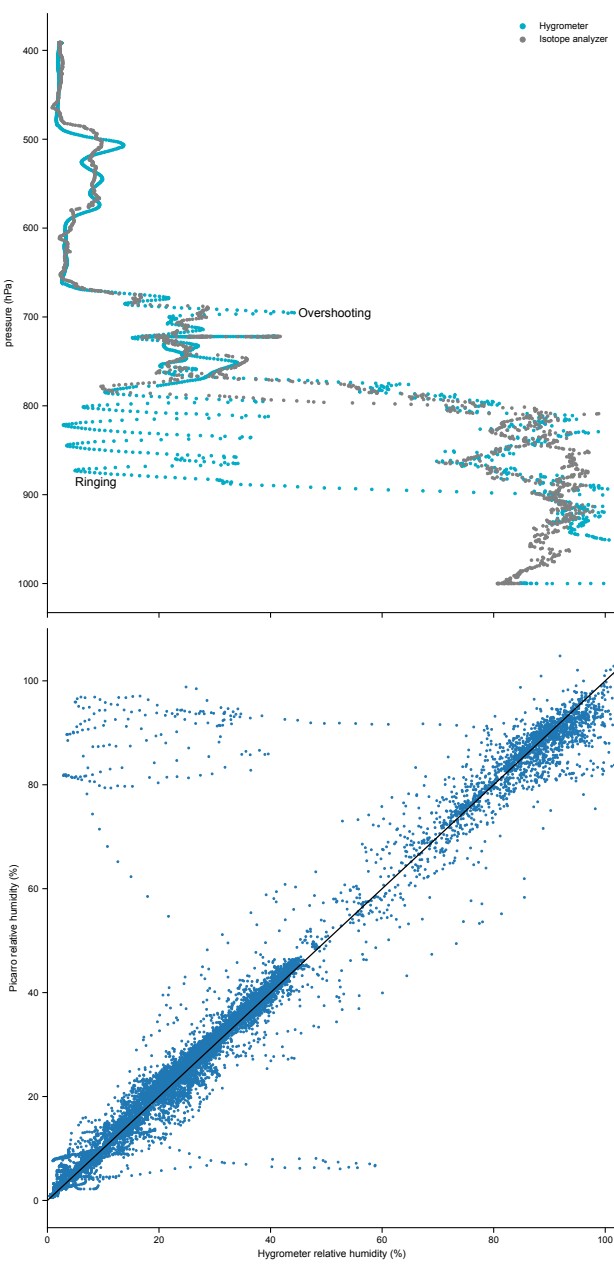

**Figure 3.** Top panel: Vertical profiles of relative humidity from the P3 hygrometer (teal) on Jan 19 2020 show overshooting, ringing, and a slow time response under the low humidity conditions found at the highest flight altitudes. These features are absent in the relative humidity profiles estimated from the water vapor isotopic analyzer (grey). Data in this panel are taken in two time windows (15:36:00 to 16:07:40 and 20:31:12 to 20:41:16 UTC) encompassing two separate slow profiles. Bottom panel: relative humidity as measured by the two sensors over the entire flight. The black line indicates equality. There is good agreement between the two water vapor sensors when relative humidity exceeds ∼20%, and when the hygrometer is not ringing or overshooting.



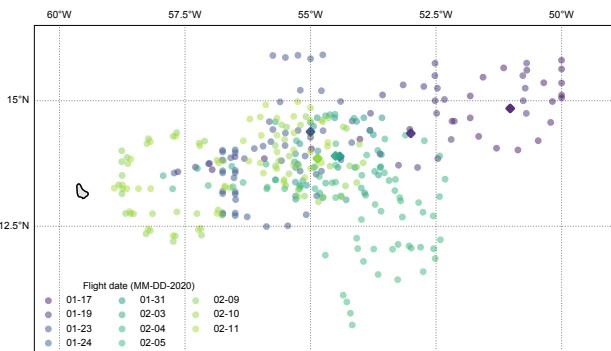

**Figure 4.** Location of dropsondes deployed during ATOMIC. Most circles (see Tab. 1) were centered on the position of the R/V Ronald H. Brown; the ships position at the start of the circle is shown with a diamond.

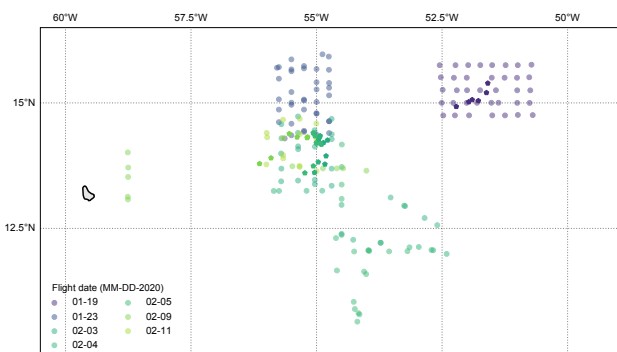

**Figure 5.** Location of ABXTs deployed during ATOMIC. Many were deployed in lawnmower patterns around the five drifting Surface Wave Instrument Floats with Tracking (SWIFT) buoys described in Quinn et al. (2020); the positions of the buoys at the mid-point of the AXBT deployment is denoted with pentagons.

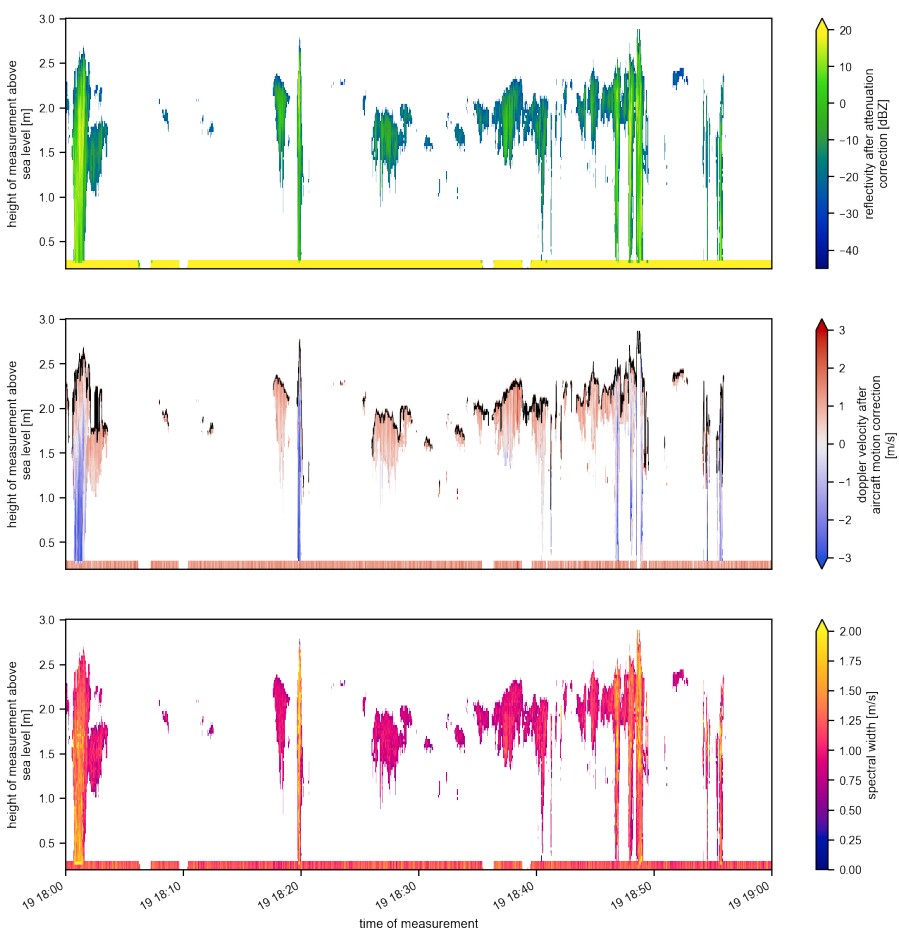

**Figure 6.** One example hour (18-19 UTC on 19 Jan 2020) of observations made by the W-band radar during ATOMIC. The top panel shows attenuation-corrected radar reflectivity ($\mathrm{dBZ}$); the middle panel the Doppler velocity after correction for aircraft motion ($\mathrm{m\,s^{-1}}$); the bottom the width of the Doppler spectrum ($\mathrm{m\,s^{-1}}$). Observations with radar signal-to-noise ratio of less than -10 $\mathrm{dB}$ have been removed for clarity.

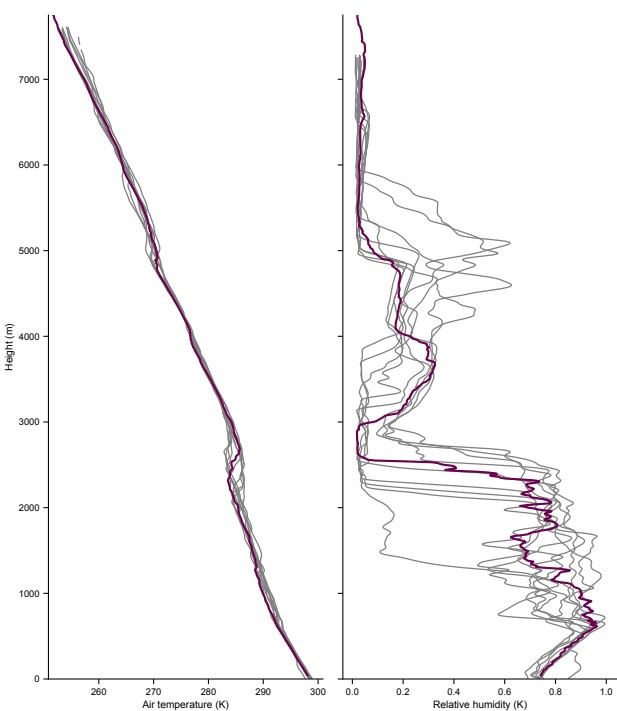

**Figure 7.** Profiles of air temperature (left) and relative humidity with respect to liquid (right) as obtained by dropsondes deployed from the P-3 (grey) during an circle made around the Ron Brown and a radiosonde launched from the ship during the same period (dark red). Dropsonde data are obtained from JOANNE (see text); radiosonde observations are obtained from the data set described in Stephan et al. (2020).



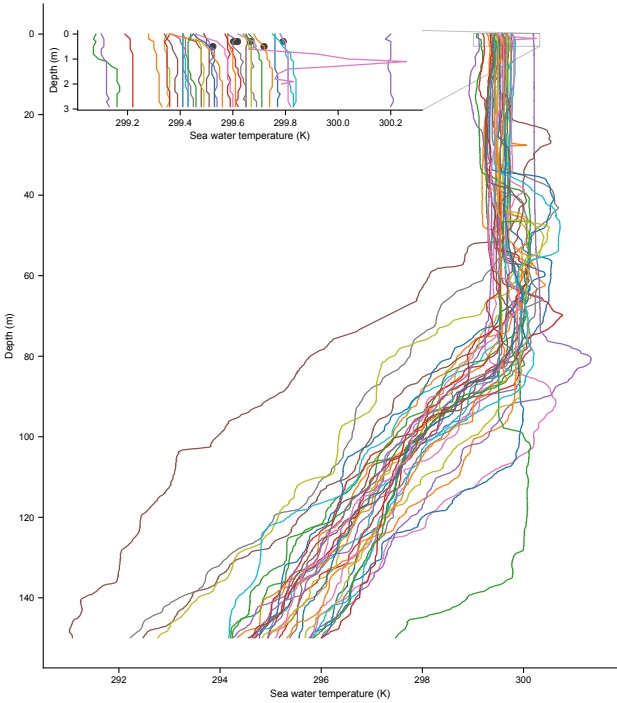

**Figure 8.** Ocean temperature profiles as measured by AXBTs deployed from the P-3 on Jan 19 2020. Data are shown between the near-surface and a depth of 150 m though the actual profiles extend to nearly 1000 meter depth. The inset shows ocean temperatures in the first few meters along with measurements from the five SWIFT buoys (see Quinn et al., 2020) surrounded by the AXBT deployments. Three of the forty AXBTs deployed did not provide valid data.

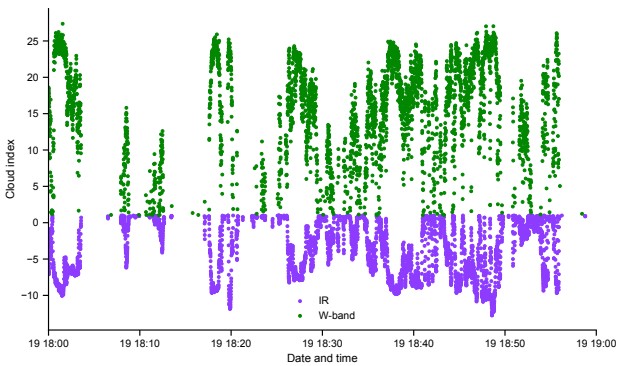

**Figure 9.** Cloud indexes based on radar ($C^{\text{radar}}$, green) and infrared radiometer ($C^{\text{IR}}$, purple) measurements for the period 18-19 UTC on 19 January (c.f. Fig. 6). Cloud is indicated by values of $C^{\text{radar}} > 0$ and $C^{\text{IR}} < 0$. Absolute values less than one have been removed for clarity. The two indexes are quite consistent with one another because clouds, when present, are typically opaque enough to be easily detectable in in both infrared and microwave measurements.

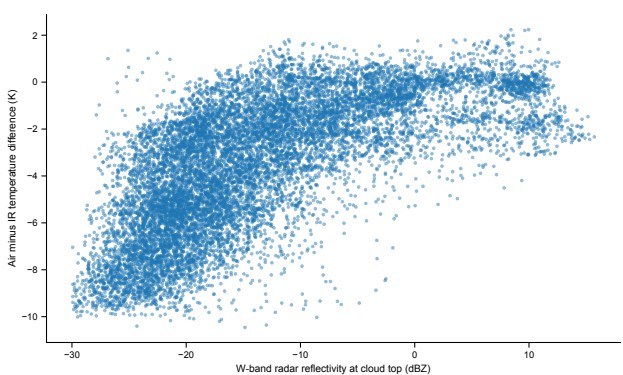

**Figure 10.** Difference between air temperature at cloud top $T_{ct}^{air}$ and the observed IR temperature $T^{IR}$ as a function of W-band radar reflectivity at cloud top for the entire flight made on 19 Jan 2020. The height of the cloud top is determined as the closest position to the observing aircraft at which the radar signal-to-noise exceeds 14 dB; air temperature as a function of height is determined from *in situ* samples made by the aircraft. Temperature differences near zero indicate that the clouds are optically thick in the infrared.





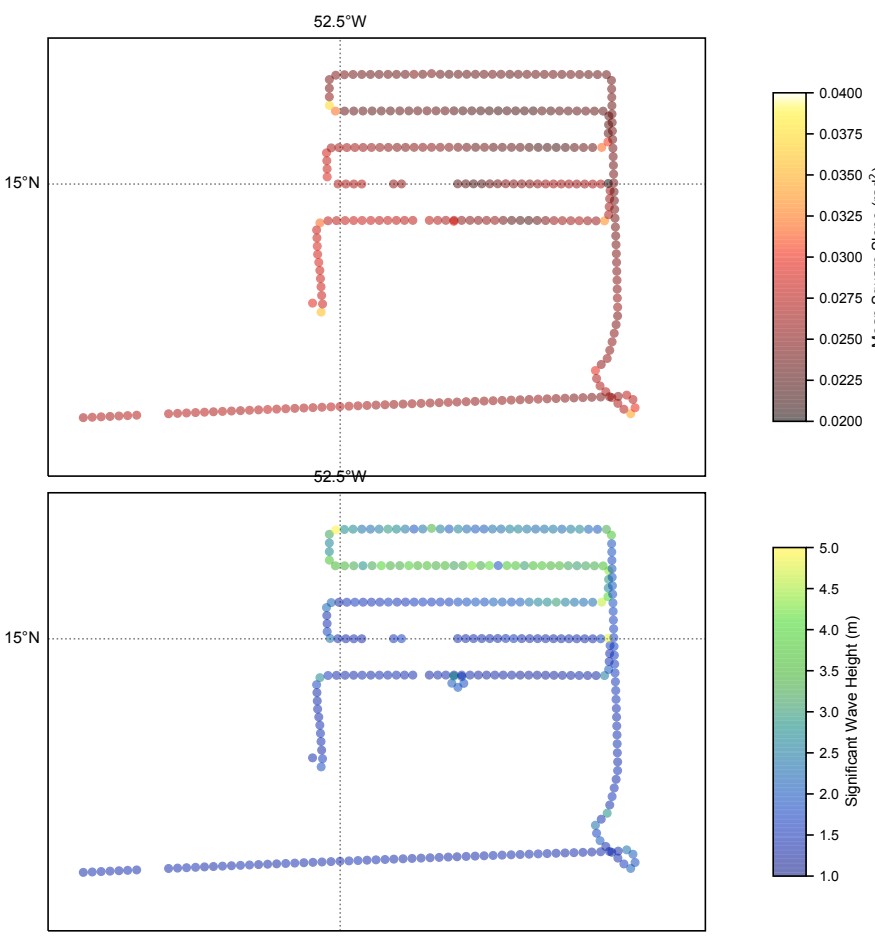

**Figure 11.** Observations of surface wave state from the Wide-Swarth Radar Altimeter (WSRA) during the P-3 flight of 19 Jan 2020. Top: Significant wave height (m). Bottom: Mean square slope ($\mathrm{rad}^2$).