# Peer review of "Observations from the NOAA P-3 aircraft during ATOMIC"

_Earth System Science Data, 2021_

## Referee Comment (RC2)

This manuscript describes measurements collected from the NOAA P-3 as part of the ATOMIC and EUREC4A field campaign aimed at studying the trade-wind cumulus environment. It seems to be one of several manuscripts describing measurement and sampling strategy from different platforms during the same field campaign.

Since this is a manuscript about a dataset, I will stick to the review guidelines of the ESSD with the following comments.

**The article:**

- The article is clearly written and provided information on all major instruments. It gives sufficient details about the sampling strategy, the instrumentation, initial data processing, post-processing for derived quantities, and discussions of data issues for the individual instruments.
- The discussions are supplemented with figures on the location and altitude of the measurements and provide some illustrations of problematic measurements such as humidity sensing.
- It might be more helpful if the article gives a short description of how the sampling strategy supports the individual objectives or hypotheses of the overall ATOMIC/EUREC4A project.

**The data and data quality:**

- All data are accessible through the provided DOI. I downloaded some examples and read the data, there is a comprehensive data description in the metadata embedded in the NetCDF data file.
- The complete dataset is a result of well-designed flight plans and in coordination with other measurement platforms in a location where trade cumulus has not been studied extensively. From this perspective, the dataset is unique and will be very useful to the scientific community although the instruments are broadly used in other data collection efforts except for isotopic analyzer and likely the W-band radar. Because of the combined air-ocean measurements through the column of the atmosphere, the operational forecast community may find a use of this dataset for coupled model evaluation as well.
- The archived dataset seems to be complete. There are Mission documents for each flight to note data quality issues, too.
- The calibration and post-processing are likely routinely used at NOAA AOC and have been evolved over the years of measurements on the standard instrument package on the N43RF (miss piggy). I would not question the methodology too much.
- Figure 3 would show better if you use different colors to denote the measurements from the two data sections. Particularly, it would allow the readers to see the comparison between the dewpoint sensor and the isotopic analyzer for each profiling period. The slow response of the dewpoint hygrometer has always been problematic at the cloud boundaries, I'm glad ATOMIC has an alternative sensor for humidity, which makes the data a lot more valuable.
- It would be nice if turbulence were part of the measurements to make the dataset even more complete. Maybe this is something to consider if future measurements are planned for a similar research topic.

Overall, I think the article is well-structured and clear. It is appropriate to support the publication of the dataset. The dataset is unique, complete, and usable in its current format and size. The metadata is also appropriate. I would recommend publication of the article with minor changes to Figure 3.

---

## Author Response (AR1)

We thank the reviewers for their generally positive reaction and appreciate the guidance as to where we were unclear. In general we have adopted reviewers' very helpful editorial suggestions without detailing each change below.

Since the manuscript was originally submitted we have finished the post-processing of the last dataset, namely the in situ cloud microphysics probes. Section 4.1.3 has therefore been heavily revised and now includes an example figure.

**Reviewer 1**

It would be helpful to increase the font size of the figure axes labels. They are a bit difficult to read, even in the electronic version.

Thanks, this was a good suggestion that we've adopted.

*Specific comments*
Figure 1. It would be helpful to increase the size of the key and if possible have larger contrasts between some of the colors. The axes are not labeled. I think some of the caption would be better placed in the text.

We have increased the size of the key as suggested. We have kept the existing color scale but now note explicitly that we have chosen a continuous set that spans the experiment so days that are close in time are also close in color. We have also kept the details in the caption as these were included to explain aspect of the figure that readers might be curious about.

Section 4.1.3. It would be good to produce the same quicklooks from all the aircraft that measured aerosols and cloud microphysics. Perhaps a note could be added to that effect in this paper.

This is a good idea which we hope to pursue but goes well beyond the scope of the paper (and requires coordination with the two other aircraft making similar measurements).

Line 237. It is difficult to see if T increases with height -- stable layer? Is it possible to provide a reason for the one outlier relative humidity profile?

By "inversion" we mean a layer of static stability, not necessarily temperature increasing with height. We now comment on the outlier dropsonde, if only to say we don't yet understand this measurement: "One of the twelve sondes shows much lower humidity from 1.5 - 2.5 km than do the others; we are assessing the all the dropsonde circles to understand if this is common or an instrumental artifact."

Section 4.2.2. It would be useful to know if the variation in temperature measured is quite normal for such a spatial scale. Provide a reference?

We have added a sentence to section 4.2.2: "Submesoscale and mesoscale eddies, fronts, and filaments in the ocean contribute to localized temperature gradients within this region (see also Fig. 4 in Quinn et al. 2021)"

**Reviewer 2**

It might be more helpful if the article gives a short description of how the sampling strategy supports the individual objectives or hypotheses of the overall ATOMIC/EUREC4A project.

An overview paper for ATOMIC, currently in preparation, will make these connections.

Figure 3 would show better if you use different colors to denote the measurements from the two data sections. Particularly, it would allow the readers to see the comparison between the dewpoint sensor and the isotopic analyzer for each profiling period...

We have distinguished the two periods in Figure 3a by using circles for one period and squares for another.

It would be nice if turbulence were part of the measurements to make the dataset even more complete

The P-3 doesn't have instrumentation devoted to, e.g., turbulent fluxes. We have added a sentence to section 4.1.1, on the post-processed flight data:  "The dataset includes measurements of vertical velocity measurements made at 1 Hz; these represent the only in situ measurements of turbulence made by the P-3.